# Body Weight as a Preferred Method for Normalizing the Computed Tomography-Derived Liver Volume in Dogs without Hepatic Disease

**DOI:** 10.3390/vetsci11040153

**Published:** 2024-03-29

**Authors:** Kosuke Kinoshita, George Moore, Masahiro Murakami

**Affiliations:** 1Department of Veterinary Clinical Sciences, College of Veterinary Medicine, Purdue University, West Lafayette, IN 47907, USA; 2Department of Veterinary Administration, College of Veterinary Medicine, Purdue University, West Lafayette, IN 47907, USA

**Keywords:** computed tomography, hepatic volumetry, canine liver disease

## Abstract

**Simple Summary:**

It is important to measure the liver size of a dog, especially when evaluating for liver disease. The common method of using radiographs can be unreliable because of the variability in the shape and size of dogs. CT hepatic volumetry can be more reliable, but to overcome the variability in dogs’ body shapes and sizes, the determination of the best normalization factor is crucial. The purpose of this study was to determine the best normalization factor for the CT-derived liver volume in dogs without liver disease. By evaluating CT scans of 41 dogs without liver disease, it was found that the body weight of a dog was the best way to normalize its CT-derived liver volume. The study suggests a simple formula: Liver volume = 19 × BW + 100; this is used to estimate the liver volume in dogs. This could be particularly helpful to veterinarians and researchers when assessing liver disease or for other scientific purposes. Further studies with larger numbers of dogs are needed to confirm these initial findings and to refine this method of measurement.

**Abstract:**

The assessment of liver size is usually performed using radiography in dogs. However, due to wide variations in patients’ sizes and body conformations, accurate diagnosis of hepatomegaly or microhepatia is difficult. Computed tomographic (CT) volumetry can quantitatively and accurately measure liver volume. However, a reliable method for the standardization or normalization of volume in dogs without hepatic disease using CT has not yet been established. The purpose of this study was to assess seven different anatomic measures for normalizing liver volume in dogs and determine the tentative range of liver volume in dogs without hepatic disease. We retrospectively searched medical records from 1 January 2017 through to 1 June 2020 and included dogs with abdominal computed tomography without hepatic disease. The liver volume, lengths of four vertebrae (T11, T12, L2, L3), diameter of the abdominal aorta, body weight, and body condition scores (BCSs) of the dogs were recorded. Forty-one client-owned dogs without evidence of hepatic disease were included. The CT-derived liver volume was 813.8 ± 326.5 cm^3^ (mean ± SD). Body weight was determined to be the most reliable single-variable method for normalizing liver volume, with a raw CT-derived liver-volume-to-body-weight ratio of 22.1 cm^3^/kg (95% CI: 12.9–31.3 cm^3^/kg) and regression prediction model of volume = 19 × BWkg + 97. However, a better normalizing factor would likely be provided by the fat-free mass if it can be accurately measured.

## 1. Introduction

The liver volume can change in dogs due to various disease processes. Generalized hepatomegaly can occur due to metabolic hepatopathies, such as steroid hepatopathy that is secondary to diabetes mellitus, hepatic lipidosis, amyloidosis, congestion, inflammation, or infiltrative round cell neoplasia [1,2]. Alternatively, a small liver size, microhepatia, may result from chronic inflammatory disease, fibrosis, cirrhosis, atrophy, or portosystemic shunts [1,2,3]. Radiography is the standard method for evaluating the liver size in dogs, where caudal displacement of the gastric axis, a rounded margin of the hepatic silhouette, and a caudal extension of the liver beyond the costal arch are used for assessment. Cranial displacement of the gastric pylorus or gastric axis is indicative of microhepatia. However, accurate radiographic diagnosis of hepatomegaly or microhepatia is challenging due to the wide variation in patients’ sizes and body conformations. To account for this variability, the radiographic size of the liver in dogs or cats is compared with the length of the vertebral body or the gastric axis relative to the rib or vertebrae, rather than using absolute measurements [4,5].

Manual computed tomographic (CT) hepatic volumetry is a noninvasive method that has recently been developed to accurately calculate the liver volume in both humans and dogs [6,7]. CT-derived estimates of liver volumes have been found to correlate closely with actual liver weights in human patients, irrespective of the etiology of chronic liver disease [7]. CT hepatic volumetry is a valuable noninvasive method for assessing liver function in both human and veterinary medicine. This technique is recommended prior to major surgical procedures in humans, such as hepatectomy, portal vein embolization, associating liver partition and portal vein ligation for staged hepatectomy, and transplantation, in order to evaluate the liver’s functional reserve [8,9]. This is important, as the liver’s functional reserve is considered one of the most important prognostic markers for the onset of liver dysfunction in the postoperative course [10,11]. The ratio of the CT-derived liver volume to the standard liver volume has been shown to correlate closely with liver function and may be a useful marker for the prognosis of cirrhosis and acute liver failure [12]. Overall, CT hepatic volumetry represents a promising approach for accurately assessing liver function and may have important implications for the management of liver disease.

In dogs, several studies using CT hepatic volumetry have demonstrated small liver volumes and postoperative increases in the liver volume normalized by total body weight in dogs with a portosystemic shunt (PSS) [13,14,15]. A previous study on liver volume, calculated by CT hepatic volumetry in dogs without hepatic disease, had a small sample size of six dogs and normalized the volume by total body weight [14]. However, there is a lack of research assessing other possible anatomic measures for normalizing liver volume, as well as information on the range of liver volumes in a larger number of dogs without hepatic disease using CT hepatic volumetry.

The purpose of this study was to assess several anatomic measurements for normalizing liver volume and provide a tentative range for CT-derived liver volumes in dogs without hepatic disease. Our hypothesis was that body weight can serve as a reliable standard for normalizing the hepatic volume, measured using CT hepatic volumetry in dogs without hepatic disease, and that CT hepatic volumetry is feasible for providing a tentative hepatic volume range in dogs without hepatic disease.

## 2. Materials and Methods

### 2.1. The Case Selection

This was a retrospective case series study. The Purdue University Veterinary Teaching Hospital (PUVTH)’s Medical Record database from 1 January 2017 through to 1 June 2020 was searched to identify dogs that had abdominal CT performed. Dogs with hepatic disease or suspected hepatic disease based on blood CBC and serum biochemistry results were excluded. All CT studies were performed using the same CT machine (Light Speed VCT, GE Medical Systems Inc., Waukesha, WI, USA). Clinical information, including breed, age, sex, body weight, and body condition score (BCS), was recorded.

### 2.2. Computed Tomographic Hepatic Volumetry

All CT studies, including entire liver, were acquired while the dogs were under general anesthesia or sedation using 64-slice third-generation CT units (Light Speed VCT, GE Medical Systems Inc., Waukesha, WI, USA) in helical scan mode. The anesthesia or sedation protocols were variable for each individual case. The scanning parameters were as follows: 120 kVp, 280 mA, and a slice thickness of 2.75 to 3.75 mm using a detailed algorithm. The images were reconstructed with a slice thickness of 5 mm. Post-contrast images were acquired at different time points after contrast administration: either a single phase at 60 s or a triple phase at 20, 55, and 95 s after the start of intravenous infusion of a nonionic iodinated contrast agent (2 mL/kg, Iohexol, Omnipaque™ 240, GE Healthcare, Marlborough, MA, USA). Pre- and post-contrast studies of the liver were reviewed by a board-certified radiologist (M. M.). Dogs with any hepatic abnormalities on the CT study were excluded from the present study. After exclusion, only pre-contrast studies were included for further CT hepatic volumetry.

CT hepatic volumetry was performed by a veterinarian (K.K.) who had received training supervised by a board-certified radiologist (M.M.) using DICOM viewer (Horos 64-bit, version 3.3.6., Purview, Annapolis, MD, USA) according to a previously published method [16]. The window width was set at 350 HU and the window level at 40 HU for all dogs. The liver segmentation was performed by manually drawing the operator-defined region of interest (ROI) on pre-contrast transverse images of the entire liver, from the cranial margin of the liver at the diaphragm to the most caudal margins of the liver adjacent to the right kidney and the spleen. The ROIs included the hepatic vessels within liver parenchyma but excluded the gallbladder and visible hepatic lobe fissures and hepatic vessels that were present outside of the hepatic parenchymal margin (Figure 1). After manual drawing of the ROIs on the hepatic parenchyma with more than 20 slices, the CT-derived liver volume was computed using the following formula to estimate the liver volume: Σ {each slice area (cm^2^) × slice thickness (cm)} × total number of slices of hepatic parenchyma/number of slices [16].

### 2.3. Variables for Evaluation

The craniocaudal vertebral lengths of T11, T12, L2, and L3, the diameter of the abdominal aorta, and the total body weight (kg) were assessed for normalization compared to the CT-derived liver volume. Vertebral length measurements were taken cranially from the T11 vertebra and caudally to the L3 vertebra. These particular vertebrae were selected because they are likely to lie in the same transverse planes as the liver and have been used to normalize liver volume in other studies [4,5,17,18]. However, the craniocaudal vertebral lengths of T13 and L1 were excluded to avoid the effect of thoracolumbar transitional vertebrae [19]. Multiplanar reconstruction images were produced to accurately measure the craniocaudal lengths of vertebrae in the longitudinal plane, and the diameter of the abdominal aorta at the level that is just cranial to the branch of the celiac artery in the transverse plane. The measurements of vertebral length and diameter of the abdominal aorta were obtained independently at three different time points.

### 2.4. Statistical Methods

Numerical variables were evaluated for normal distribution using the Shapiro–Wilk test. Summary statistics are presented as mean ± SD for variables that did not violate the assumptions of normality, and as median (range) for age, weight, and BCS which had nonparametric distributions. The assessment of variables as candidates for normalization of CT-derived liver volume was performed in two ways. First, the correlation between the liver volume and the craniocaudal vertebral lengths of T11, T12, L2, and L3, and the diameter of the aorta was assessed by the Pearson product–moment correlation coefficient (*r*); the correlation of liver volume with total body weight and body condition score (BCS) was assessed by the Spearman correlation coefficient (*rho*). Then, linear regression was performed to analyze the multivariable interrelationships between the craniocaudal vertebral lengths of T11, T12, L2, and L3, diameter of the aorta, and total body weight as potential normalized indices of liver volume in dogs. To assess for potential confounding between independent variables, stepwise backwards regression was used to identify coefficient changes of >20% with the subtraction/addition of variables. Prior to accepting the analyses, assumptions of linearity, homoscedasticity, and normality of residuals were checked and confirmed. Commercially available software (STATA SE, v.18.0, StataCorp LLC, College station, TX, USA) was used to conduct all analyses. A statistically significant difference was indicated by a *p*-value of 0.05.

## 3. Results

A total of 41 dogs were included in the study, and their clinical histories and laboratory test results were evaluated. The dogs had a body weight at CT scan ranging from 6.5 kg to 84.2 kg (median 37.6 kg) and an age ranging from 1 year to 15 years (median 7.1 years), with a BCS ranging from 3 to 8 (median 5). The breeds included two Belgian Malinois, two French Bulldogs, four German Shepherd Dogs, five Golden Retrievers, six Labrador Retrievers, eight Mixed-Breed Dogs, two Rottweilers, and one each of the following breeds: Alaskan Malamute, Belgian Sheepdog, English Bulldog, German Shorthaired Pointer, Goldendoodle, Great Dane, Havanese Terrier, Jack Russel Terrier, Mastiff, Bull Mastiff, Old English Mastiff, and Saint Bernard. The mean (±SD) CT-derived liver volume for all dogs was 813.8 ± 326.5 cm^3^ (median 779 cm^3^; range 195 to 1598 cm^3^) (Table 1).

Body weight showed the strongest correlation with the CT-derived liver volume among the assessed anatomic measurements (*rho* = 0.78, *p* < 0.001), although the other five variables did have moderate correlations with liver volume (*r* = 0.69–0.76; *p* < 0.001) (Figure 2).

The multivariable analysis revealed that total body weight was the only factor that showed a significant association with the CT-derived liver volume (*p* < 0.01). The other factors that were included in the model (craniocaudal vertebral length of T11, T12, L2, and L3; diameter of the aorta; and BCS) were not significantly associated with the CT-derived liver volume (*p* > 0.13). Confounding, however, was suggested to occur by a change in the body weight’s multivariable regression coefficient from 15.65 (95% CI: 9.80–21.49) to 18.96 (95% CI: 15.82–22.09) in univariate regression; this change was primarily due to the inclusion/exclusion of the L3 length in the model.

The raw mean liver-volume-to-body-weight ratio was 22.1 ± 4.6 cm^3^/kg (95% CI: 12.9–31.3). The use of this simple 22-fold ratio for normalization showed potential bias compared to the regression model options with one, two, or seven dependent variables, with potential underestimation of volume in smaller (<20 kg) dogs and potential overestimation in larger (>55 kg) dogs (Figure 3). In spite of the statistical confounding, the BW with L3 fitted regression model was not significantly different to the simpler univariate model of Volume = 19 × BW + 97.

The BCS showed a small negative association with the CT-derived liver-volume-to-body-weight ratio (rho = −0.40; *p* = 0.012) (Figure 4).

## 4. Discussion

In the present study, we measured the CT-derived liver volumes in forty-one dogs without hepatic disease. A previous study had assessed liver volume using CT hepatic volumetry in 36 small dogs without liver disease, with a mean body weight of 5.42 ± 3.09 kg (mean ± SD) [17]. However, that study only included dogs with a radiographically normal liver size. The CT-derived liver volume that was reported for small dogs in that study was 167 ± 111.88 cm^3^ (mean ± SD) [17]. In contrast, our study used CT hepatic volumetry to assess dogs of various sizes without hepatic disease, ranging from approximately 10 to 75 kg in body weight. The CT-derived liver volume in our study showed a large SD, indicating a large variation. Dogs exhibit wide variation in body conformation due to factors such as breed and individual variability, making it challenging to estimate their liver size based solely on their liver volume.

To establish a normal liver volume reference using CT hepatic volumetry in dogs, a reliable internal control for normalization is crucial. Previous studies have demonstrated that the total body weight can be used to normalize the liver volume in dogs, as it has a strong positive correlation with the liver volume [13,14,15,17]. In one study, multiple linear regression analysis revealed that the CT-derived liver volume was strongly and positively correlated with the total body weight and moderately-to-strongly correlated with radiographic parameters, supporting the use of the total body weight for optimal normalization [17]. However, CT indices were not evaluated as normalization indices for liver volume in that study. In the present study, the primary purpose was to establish the most reliable internal control to normalize the liver volume in dogs of various sizes. The present study compared the CT-derived liver volume with other internal controls that are used in a CT study, including the craniocaudal vertebral length of T11, T12, L2, and L3 and the diameter of the aorta. The multiple linear regression analysis showed that only total body weight showed statistical significance, with a strong positive correlation with the CT-derived liver volume. The coefficient of variation of the liver volume normalized by total body weight was the smallest compared to other internal controls evaluated in the present study.

These findings are consistent with previous studies of CT hepatic volumetry that is normalized by total body weight in dogs. One study on six dogs without hepatic disease found that the CT-derived liver-volume-to-total-body-weight ratio was 24.5 ± 5.6 cm^3^/kgBW [14]. Another study on small dogs with radiographically normal liver sizes found that the CT-derived liver volume normalized by total body weight was 30.3 ± 6.1 cm^3^/kg (mean ± SD) [17]. In a study evaluating changes in liver volume in dogs with PSS before and after surgery using CT hepatic volumetry, a significant increase in the postoperative liver-volume-to-total-body-weight ratio was found, with postoperative liver volume of 29.0 ± 5.1 mL/kgBW (mean ± SD) [15]. These results were similar to the present study’s results of a CT-derived liver volume of 22.1 ± 4.6 cm^3^/kgBW (mean ± SD) with a 95% confidence interval of 12.9 to 31.3 cm^3^/kgBW. The present study confirms that total body weight is the most reliable internal control for normalizing the CT-derived liver volume in dogs of various sizes. Thus, the provided values of CT-derived liver volume normalized by total body weight are likely to be reliable normalized liver volumes in dogs without hepatic disease. However, due to the small number of dogs included in the present study, we report these values as tentative ranges of normalized CT-derived liver volumes in dogs without hepatic disease.

Assessing the liver volume is important in human medicine for two key reasons: determining the size of a graft for liver transplantation [20,21] and estimating hepatic drug clearance using pharmacokinetic models [22,23]. The body surface area (BSA) is commonly used to calculate the liver volume for both of these purposes [24,25,26,27], and one meta-analysis study found that it correlates better with the liver volume than body weight does [28]. Additionally, BSA-based estimates of liver volume have been shown to be accurate even in children with varying body conditions [29]. In veterinary medicine, the BSA is commonly used to calculate chemotherapeutic drug doses for oncology patients [30,31]. This method helps account for the effects of body condition and estimate the true metabolic activity of the body, which is likely to correlate with the liver volume in patients without liver disease. To calculate the BSA in dogs, a formula that takes body weight and body height into account is typically used. However, in the present study, the authors were unable to collect body height data due to the retrospective nature of the study. Instead, we used the body condition score (BCS) as a contributor to the BSA. The present study found that the CT-derived liver-volume-to-body-weight ratio was negatively associated with the BCS. In dogs with low body condition scores, body weight affected the normalizing factor too much, making the normalized CT-derived liver volume larger than the average, contrary to the high body condition score dogs. Dogs with low body condition scores should have relatively larger fat-free true metabolic masses, and high body condition score dogs will have relatively lower fat-free true metabolic masses. Thus, using the fat-free true metabolic body mass as an internal control for normalizing the CT-derived liver volume may cancel out the effect of the negative association between the BCS and normalized CT-derived liver volume, becoming a better internal control for normalization. However, accurately evaluating this variable can be difficult, so total body weight is more applicable in a clinical setting. Additionally, it is important to consider the possibility of underestimating the calculated CT-derived liver-volume-to-body-weight ratio in dogs with higher BCSs.

There are several limitations to the present study. Firstly, the relatively small sample size limits the establishment of a normal reference range for normalized CT-derived liver volume in dogs. This may not represent the broader dog population, as more than 120 cases are recommended to establish a reference interval in veterinary medicine [32]. Consequently, we have reported a tentative range of normalized CT-derived liver volume values. Although the results of the present study offer useful preliminary data, further research involving a larger number of cases is necessary to provide a more robust reference range for normalized CT-derived liver volume in dogs.

Another limitation is the absence of liver biopsy or necropsy data to confirm the absence of liver disease in the dogs who were included in the study. This may potentially introduce bias if any of the dogs had undetected liver disease. However, due to ethical considerations and the invasive nature of these procedures, obtaining such confirmatory data in a healthy population is impractical. Therefore, relying on comprehensive clinical histories and laboratory test results represents the best available approach to ensure that the dogs included in the study were free from hepatic disease.

Finally, the retrospective nature of the study limited the ability to collect certain data, notably body height, which could have been useful in calculating the body surface area (BSA) for a potentially more accurate internal control for normalizing the liver volume. The use of the body condition score (BCS) as a surrogate for the BSA has its limitations, as it does not fully account for body conformation variability among different breeds.

## 5. Conclusions

The results of this study indicate that using a CT-derived liver volume alone is not sufficient for accurately evaluating the liver size in dogs due to the wide range of body sizes and conformations. We found that the total body weight is the most effective internal control for normalizing the liver volume. In dogs without hepatic disease, the CT-derived liver-volume-to-body-weight ratio was calculated as 22.1 ± 4.6 (mean ± SD). A simplified univariate model given by the equation Liver Volume = 19 × BW + 100 can be used to estimate the approximate liver volume. However, it is important to note that the fat-free mass might potentially serve as a more effective internal control than the total body weight, considering the negative association between the liver-volume-to-body-weight ratio and the BCS. Further research is needed to examine CT hepatic volumetry that is normalized with the BSA or fat-free mass, which may enhance the accuracy of CT hepatic volumetry assessments in clinical cases. Additionally, a normal reference range for normalized CT-derived liver volumes using a larger sample of dogs without hepatic disease would be beneficial.

## Figures and Tables

**Figure 1 vetsci-11-00153-f001:**
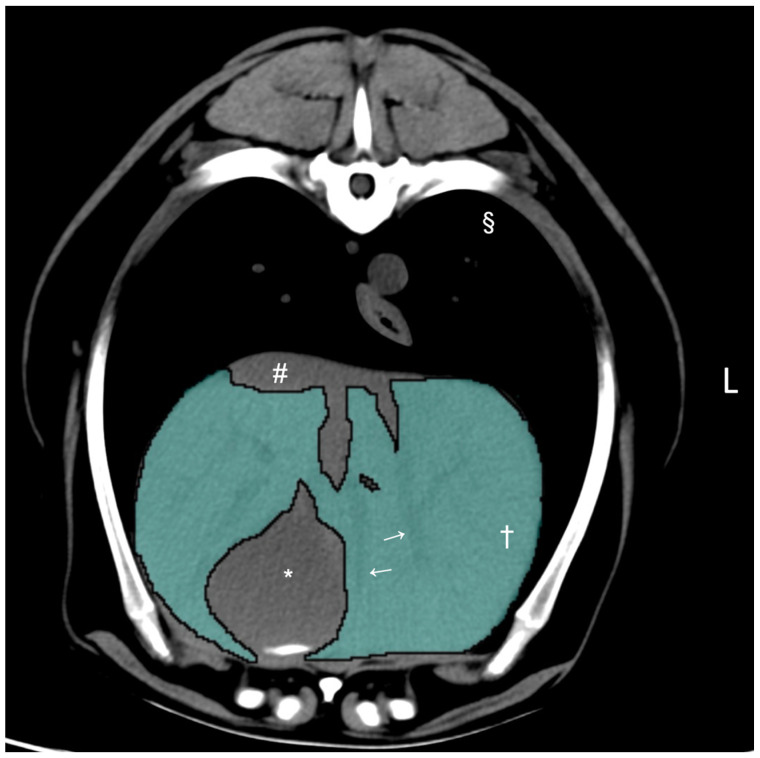
Pre-contrast transverse abdominal CT images used for CT hepatic volumetry in dogs. The segmentation of the liver was manually selected as the region of interest (ROIs: highlighted in blue). Note that the hepatic vessels within the liver parenchyma were included (white arrows). The gallbladder, hepatic lobe fissure, and hepatic vessels present outside the hepatic parenchyma were excluded. Window width, 350 HU, window level, 40 HU. Liver parenchyma (†), gallbladder (*), caudal vena cava (#), and pulmonary parenchyma (§).

**Figure 2 vetsci-11-00153-f002:**
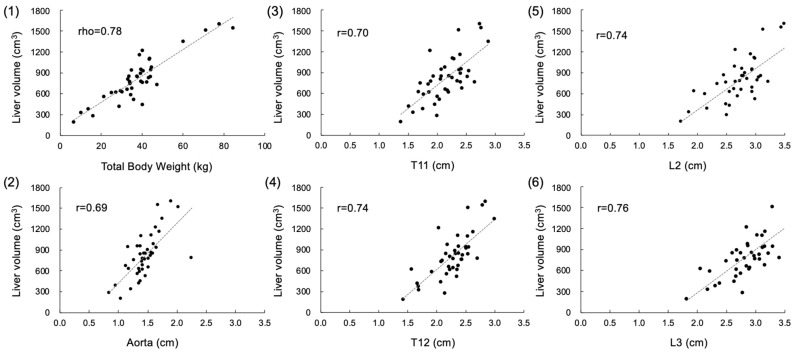
Scatter plots of CT-derived liver volume (*y*-axis) and anatomic measures of (**1**) total body weight, (**2**) diameter of abdominal aorta, (**3**) craniocaudal length of T11, (**4**) craniocaudal length of T12, (**5**) craniocaudal length of L2, and (**6**) craniocaudal length of L3 in normal dogs. Linear regression line and correlation coefficient are shown in each graph.

**Figure 3 vetsci-11-00153-f003:**
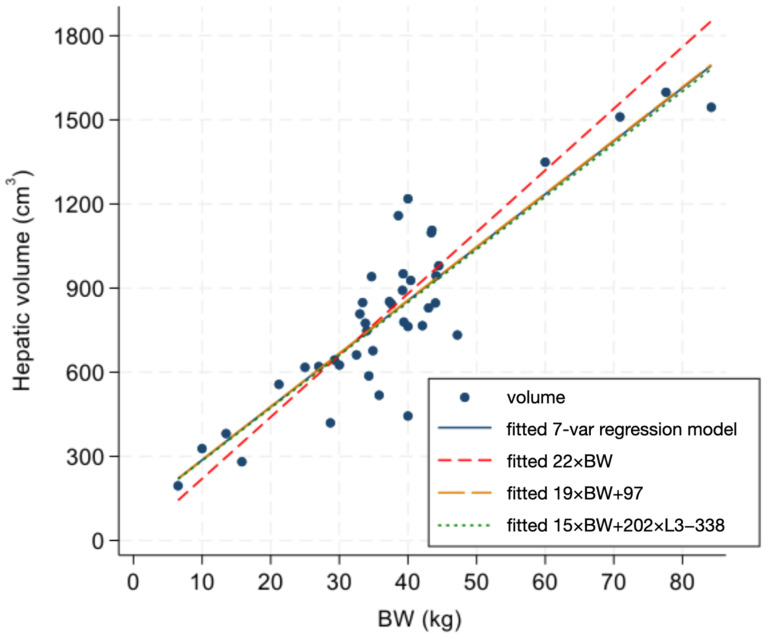
Scatter plot of CT-derived hepatic volume and body weight (BW) in 41 dogs with fitted regression lines from 4 different regression models. Dependent variable (volume) and 7 independent variables are shown in Table 1.

**Figure 4 vetsci-11-00153-f004:**
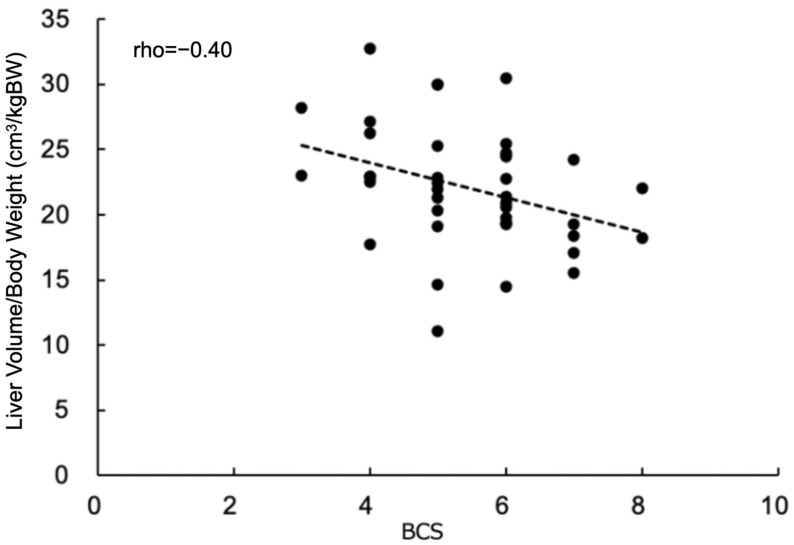
Correlation between liver volume normalized by total body weight and body condition score (BCS). Regression line and correlation coefficient are shown in the graph.

**Table 1 vetsci-11-00153-t001:** Summary statistics of liver volume and other variables.

Variable	Mean ± SD	Median (Range)
Liver volume (cm^3^)	813.8 ± 326.5	779.0 (194.9–1598.5)
T11 (cm)	2.14 ± 0.35	2.1 (1.4–2.9)
T12 (cm)	2.28 ± 0.34	2.3 (1.4–3.0)
L2 (cm)	2.75 ± 0.40	2.8 (1.7–3.5)
L3 (cm)	2.87 ± 0.41	2.9 (1.8–3.6)
Aorta (cm)	1.46 ± 0.27	1.4 (0.8–2.3)
BW (kg)	nonparametric	37.7 (6.5–84.2)
BCS	nonparametric	5 (3–8)

Abbreviations: T11, craniocaudal length of 11th thoracic vertebra; T12, craniocaudal length of 12th thoracic vertebra; L2, craniocaudal length of 2nd lumber vertebra; L3, craniocaudal length of 3rd lumber vertebra; Aorta, diameter of aorta; BW, body weight; BCS, body condition score [9-point scale].

## Data Availability

The data that support the findings of this study are available from the corresponding author upon reasonable request.

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
