# Peer review of "Body Weight as a Preferred Method for Normalizing the Computed Tomography-Derived Liver Volume in Dogs without Hepatic Disease"

_vetsci, 2024, doi:10.3390/vetsci11040153_

Round 1

Reviewer 1 Report

Comments and Suggestions for Authors

Dear Authors,

I reviewed the manuscript entitled "Body weight as a preferred method for normalizing CT-derived liver volume in dogs without hepatic disease"

The manuscript is well written, the study is well organized and the topic is interesting, since liver size is very difficult to assess in veterinary diagnostic imaging. The study describes the assessment of 7 different anatomic measures for normalizing liver volume in dogs and determine the  range of liver volume in dogs without hepatic disease.

I have only some minor comments (see below), therefore I recommend minor revision.

Line 100: Why did the Authors use for volumetry only non-contrast images?

line 130: Was abdominal aorta size calculated in non-contrast images?

line 154: the age range was reported to be 1-15 years. Did the Authors notice a correlation between normalized liver volume and age? Please discuss the possible correlation of age ad liver volume, as it is described in humans (K Harada et al2021).

Author Response

March 13, 2024

Subject: Revision and resubmission of manuscript ID vetsci-2880744 to Veterinary Sciences.

Dear reviewer 1,

Thank you for reviewing our manuscript, and the opportunity to revise our manuscript entitled "Body weight as a preferred method for normalizing CT-derived liver volume in dogs without hepatic disease". The suggestions offered have been immensely helpful, and we also appreciate your insightful comments on revising the manuscript.

I have included your comments immediately after this letter and responded to them individually, indicating exactly how we addressed each concern or problem and describing the changes we have made. The revisions have been approved by all authors. The changes are highlighted by using bold in the reply, and the revised manuscript will be resubmitted with this letter.

Our response follows (the reviewers’ comments are in italics).

Reviewer: 1

Dear Authors,

I reviewed the manuscript entitled "Body weight as a preferred method for normalizing CT-derived liver volume in dogs without hepatic disease"

The manuscript is well written, the study is well organized and the topic is interesting, since liver size is very difficult to assess in veterinary diagnostic imaging. The study describes the assessment of 7 different anatomic measures for normalizing liver volume in dogs and determine the range of liver volume in dogs without hepatic disease.

I have only some minor comments (see below), therefore I recommend minor revision.

Line 100: Why did the Authors use for volumetry only non-contrast images?

Reply:

Thank you for your positive feedback and for raising this important question. The decision to use non-contrast images for volumetric analysis was made to avoid the risk of blooming artifacts that can occur with post-contrast images and potentially interfere with accurate delineation of organ margins. In this study, the non-contrast images provided sufficient clarity to accurately identify the liver margins, supporting our contention that they provide more reliable measurements of organ size in the absence of contrast enhancement when the liver margins are identifiable.

line 130: Was abdominal aorta size calculated in non-contrast images?

Reply:

Abdominal aortic dimensions were also assessed using non-contrast images. Consistent with our methodological approach, all measurements for the study were derived from non-contrast images to ensure uniformity of imaging conditions for all assessed parameters.

line 154: the age range was reported to be 1-15 years. Did the Authors notice a correlation between normalized liver volume and age? Please discuss the possible correlation of age ad liver volume, as it is described in humans (K Harada et al · 2021).

Reply:

We appreciate your question regarding the correlation with age. Although the primary objective of our study was to identify the optimal normalization factor for CT liver volumetry, we recognize the importance of age as a variable in liver volume assessment. However, the relationship between age and normalized liver volume was beyond the scope of our current study. However, we recognize its importance as highlighted in the human literature (K Harada et al. 2021). In fact, we are currently compiling a larger dataset to establish a normal reference range of CT-derived liver volumes in dogs without liver disease, and within this study, we aim to thoroughly explore the relationship between normalized liver volume and other factors, including age, in a forthcoming publication.

We hope the revised manuscript will better suit Veterinary Sciences, and we thank you for your continued interest in our manuscript.

Sincerely,

Masahiro Murakami

Purdue University College of Veterinary Medicine

Department of Veterinary Clinical Sciences
West Lafayette, IN 47907.

Telephone: +1-(765)-494-1107

Reviewer 2 Report

Comments and Suggestions for Authors

Brief summary:

The main objective of this article is to determine the size of the healthy liver by measuring its volume with computed tomography and normalizing it according to the size and shape of the dog. After studying several variables, it is stated that the weight of the animal is the best variable for normalize the liver volumen calculation in different breeds but it is concluded that CT liver volume alone is not sufficient to accurately assess liver size in dogs, due to the wide range of body sizes and conformations.

The main strength of this study is that it provides unpublished information on the relationship between liver volume and various anatomical measures to correct volume for animal size and lays the foundation for using these liver volume measures for clinical medical and/or surgical protocols.

General Comments:

The standardization of liver lesions and pathologies and the diagnosis of liver diseases is still a gap in the scientific literature. There is a lack of scientific evidence on the correlation between liver size and liver functionality and the relationship with various pathologies, but this article can provide information to move in that direction.

The main weakness of this article is that it is not possible to extrapolate the results from healthy animals to diseased animals, with variations in body weight resulting from their own liver pathologies. But more studies are needed, as the authors state.

It should explain why these anatomical variables were used for assessment and not others. There is a lack of bibliographical references to justify the use of these references. For example, that they are used in conventional radiology or others; this could justify why T13 or L1 were not used?

The statistical analysis seems to be correct to the best of my knowledge. The manuscript includes the limitations to which the results obtained are subject.

The results are in line with the proposed objectives and the methodology used is adequate to achieve them. There is therefore a traceability between the conclusions and the data and study carried out.

Specific comments 

Authors must describe the anesthetic protocol used and also indicate the contrast media used and the dosage in order to complete the information appropriately in material and methods.

Author Response

March 13, 2024

Subject: Revision and resubmission of manuscript ID vetsci-2880744 to Veterinary Sciences.

Dear reviewer 2,

Thank you for reviewing our manuscript, and the opportunity to revise our manuscript entitled "Body weight as a preferred method for normalizing CT-derived liver volume in dogs without hepatic disease". The suggestions offered have been immensely helpful, and we also appreciate your insightful comments on revising the manuscript.

I have included your comments immediately after this letter and responded to them individually, indicating exactly how we addressed each concern or problem and describing the changes we have made. The revisions have been approved by all authors. The changes are highlighted by using bold in the reply, and the revised manuscript will be resubmitted with this letter.

Our response follows (the reviewers’ comments are in italics).

Reviewer: 2

Reviewer 2

Brief summary:

The main objective of this article is to determine the size of the healthy liver by measuring its volume with computed tomography and normalizing it according to the size and shape of the dog. After studying several variables, it is stated that the weight of the animal is the best variable for normalize the liver volumen calculation in different breeds but it is concluded that CT liver volume alone is not sufficient to accurately assess liver size in dogs, due to the wide range of body sizes and conformations.

The main strength of this study is that it provides unpublished information on the relationship between liver volume and various anatomical measures to correct volume for animal size and lays the foundation for using these liver volume measures for clinical medical and/or surgical protocols.

General Comments:

The standardization of liver lesions and pathologies and the diagnosis of liver diseases is still a gap in the scientific literature. There is a lack of scientific evidence on the correlation between liver size and liver functionality and the relationship with various pathologies, but this article can provide information to move in that direction.

The main weakness of this article is that it is not possible to extrapolate the results from healthy animals to diseased animals, with variations in body weight resulting from their own liver pathologies. But more studies are needed, as the authors state.

It should explain why these anatomical variables were used for assessment and not others.

There is a lack of bibliographical references to justify the use of these references. For example, that they are used in conventional radiology or others; this could justify why T13 or L1 were not used?

Reply:

We appreciate your constructive feedback on the manuscript. In response to your concerns regarding the selection of vertebral reference points, we have revised the manuscript to include additional references that justify our methodology.

Line 130-135:

Vertebral length measurements were taken cranially from the T11 vertebra and caudally to the L3 vertebra. These particular vertebrae were selected because they are likely to lie in the same transverse planes as the liver and have been used to normalize liver volume in other studies[4,5,17,18]. However, the craniocaudal vertebral lengths of T13 and L1 were excluded to avoid effect of thoracolumbar transitional vertebrae[19].

The statistical analysis seems to be correct to the best of my knowledge. The manuscript includes the limitations to which the results obtained are subject.

The results are in line with the proposed objectives and the methodology used is adequate to achieve them. There is therefore a traceability between the conclusions and the data and study carried out.

Specific comments 

Authors must describe the anesthetic protocol used and also indicate the contrast media used and the dosage in order to complete the information appropriately in material and methods.

Reply:

We appreciate your request for additional details on the anesthesia and sedation protocols and the use of contrast media. Given the retrospective nature of our study, the anesthetic and sedation protocols were not consistent across cases. We have now included a statement in the manuscript to clarify this variability.

Regarding contrast media, we have amended our methodology section to include the specific contrast media used, its dosage, and the timing of imaging following administration.

Line 95-102:

The anesthesia or sedation protocols were variable for each individual case. The scanning parameters were as follows: 120 kVp, 280 mA, and a slice thickness of 2.75 to 3.75 mm using a detailed algorithm. The images were reconstructed with a slice thickness of 5 mm. Post-contrast images were acquired at different time points after contrast administration: either a single phase at 60 seconds or a triple phase at 20, 55, and 95 seconds after the start of intravenous infusion of a nonionic iodinated contrast agent (2 mL/kg, Iohexol, Omnipaque™240, GE Healthcare, Marlborough, MA, USA).

We hope the revised manuscript will better suit Veterinary Sciences, and we thank you for your continued interest in our manuscript.

Sincerely,

Masahiro Murakami

Purdue University College of Veterinary Medicine

Department of Veterinary Clinical Sciences
West Lafayette, IN 47907.

Telephone: +1-(765)-494-1107
